# Aim24 and MICOS modulate respiratory function, tafazzin-related cardiolipin modification and mitochondrial architecture

**Max Emanuel Harner[1], Ann-Katrin Unger[2], Toshiaki Izawa[1], Dirk M Walther[1], Cagakan Özbalci[3], Stefan Geimer[2], Fulvio Reggiori[4], Britta Brügger[3], Matthias Mann[1], Benedikt Westermann[2], Walter Neupert[1]***

[1]Max Planck Institute of Biochemistry, Martinsried, Munich, Germany; [2]Cell Biology and Electron Microscopy, University of Bayreuth, Bayreuth, Germany; [3]Heidelberg University Biochemistry Centre, Heidelberg, Germany; [4]Department of Cell Biology, University Medical Centre Utrecht, Utrecht, Netherlands

**Abstract** Structure and function of mitochondria are intimately linked. In a search for components that participate in building the elaborate architecture of this complex organelle we have identified Aim24, an inner membrane protein. Aim24 interacts with the MICOS complex that is required for the formation of crista junctions and contact sites between inner and outer membranes. Aim24 is necessary for the integrity of the MICOS complex, for normal respiratory growth and mitochondrial ultrastructure. Modification of MICOS subunits Mic12 or Mic26 by His-tags in the absence of Aim24 leads to complete loss of cristae and respiratory complexes. In addition, the level of tafazzin, a cardiolipin transacylase, is drastically reduced and the composition of cardiolipin is modified like in mutants lacking tafazzin. In conclusion, Aim24 by interacting with the MICOS complex plays a key role in mitochondrial architecture, composition and function.

**\*For correspondence:** Neupert@
biochem.mpg.de

**Competing interests:** The authors declare that no competing interests exist.

**Reviewing editor**: Richard J Youle, National Institute of Neurological Disorders and Stroke, National Institutes of Health, United States

## Introduction

Research on mitochondria in the previous decades has been focused on their role in oxidative phosphorylation and related metabolic processes, on their genetics, dynamics, biogenesis, in programmed cell death and in diseases. While significant progress was made in these fields, relatively little attention was devoted to the internal organization of mitochondria. Early electron microscopic studies had revealed the highly elaborate architecture and its large variety in mitochondria of different species, tissues and cells (*Fawcett, 1981*). Few cohesive attempts were made to elucidate the genes and other factors responsible for the architecture of the mitochondria. It is now becoming evident that in order to understand the communication and regulation between mitochondria, cytosol and nucleus this area of mitochondrial biology must be strengthened.

Metabolism, biogenesis, dynamics and inheritance of mitochondria are intimately linked to their architecture (*Mannella et al., 2001*; *John et al., 2005*; *Sun et al., 2007*; *Mannella, 2008*; *Hess et al., 2009*; *Perkins et al., 2010*; *Harner et al., 2011*; *Perkins and Ellisman, 2011*). The internal organization of the mitochondria plays also an important role in apoptosis and mitophagy (*Nowikovsky et al., 2007*; *Sun et al., 2007*; *Merkwirth et al., 2008*; *Rolland et al., 2009*). Mitochondrial architecture is altered in many diseases affecting the performance of the organelle (*Zick et al., 2009*). Thus, it is essential to identify and characterize the proteins and lipids responsible for mitochondrial architecture with regard to structure, suborganellar location, function and dynamics.

**eLife digest** Respiration is vital to all living things because it is the process by which nutrients are converted into useful energy. Within our cells, organelles called mitochondria harness this energy and store it within molecules of ATP. This energy can then be released, when it is needed, by breaking down an ATP molecule into simpler chemicals.

When viewed under an electron microscope, it can be seen that an individual mitochondrion has two membranes: a highly folded inner membrane, and an outer membrane that is fairly smooth. However, some of the genes, proteins and other molecules that determine the complex shapes of mitochondria have not been identified.

Using a technique called mass spectrometry, a protein called Aim24 was previously shown to accumulate at the sites where the two membranes of a mitochondrion make contact with each other. Now, Harner et al. have discovered that Aim24 proteins are transported into mitochondria and embedded within the inner membranes. Aim24 is needed to assemble and maintain the stability of the protein complex that anchors the inner membrane to the outer membrane: mitochondria that lack the Aim24 protein have an irregular structure.

Furthermore, Harner et al. showed that Aim24 is needed to form other protein complexes—also within the inner membrane—that work together to harness the energy that is released when nutrients are broken down, so that it can be stored as ATP. Cells that lack the Aim24 gene also showed changes in the kinds of molecules found within the membranes of the mitochondria.

This work shows how difficult it can be to determine how the shapes of structures within cells, such as mitochondria, are controlled—as one protein can have many roles that cannot be easily teased apart. The on-going challenge is to uncover how different proteins, and other molecules in the membranes of mitochondria, interact to determine the structure, and consequently function, of the mitochondria in a cell.

Mitochondria are separated from the cytosol by a mitochondrial envelope consisting of the proximate outer membrane (OM) and the inner boundary membrane (IBM). The cristae protrude into the matrix from narrow tubule-like or slot-like crista junctions (CJ) at which the membrane has a strong negative curvature. In most cases they form leaf-like or tubule-like structures. At their edges, the crista rims, the membrane is bent with a strong positive curvature (*Rabl et al., 2009*).

A considerable number of proteins have been identified which influence mitochondrial architecture. To some of them distinct functions could be assigned. These proteins include the subunits of the $F_1F_O$-ATP synthase that are responsible for oligomerisation of this enzyme complex, such as subunits e, g and 4 (*Arnold et al., 1998*; *Rabl et al., 2009*; *Davies et al., 2012*). These subunits apparently introduce positive curvature into the dimers and oligomers of the $F_1F_O$-ATP synthase and thereby into the membrane which then leads to the formation of crista rims (*Dudkina et al., 2005*; *Strauss et al., 2008*; *Rabl et al., 2009*). Mitofilin is another protein whose architectural role was recognized early on. It is required for the formation of crista junctions (*John et al., 2005*). Later, it was proved to be a component of a larger complex, the MICOS/MINOS/Mitos complex (*Harner et al., 2011*; *Hoppins et al., 2011*; *von der Malsburg et al., 2011*). An agreement between researchers in this field was recently reached to introduce a uniform nomenclature (*Pfanner et al., 2014*). The MICOS/MINOS/Mitos complex is now called MICOS ('mitochondrial contact site and cristae organizing system'). MICOS is composed of at least six subunits whose names are now (former names in brackets): Mic60 (Fcj1), Mic10 (Mcs10/Mos1/Mio10), Mic12 (Mcs12/Aim5), Mic19 (Mcs19/Aim13), Mic26 (Mcs29/Mos2, Mio27) and Mic27 (Mcs27/Aim37). All subunits are integrated in or associated with the inner membrane. They contact the outer membrane by interacting with the TOB/SAM complex and the Ugo1-Fzo1 complex. Thereby they anchor MICOS and the crista junctions to the outer membrane (*Harner et al., 2011*). The MICOS complex, or in some cases components of the complex, have been shown to exist also in higher organisms (*Xie et al., 2007*; *Park et al., 2010*; *Darshi et al., 2011*; *Head et al., 2011*; *Alkhaja et al., 2012*; *An et al., 2012*; *Ott et al., 2012*; *Weber et al., 2013*).

Further outer and inner membrane proteins with an important role in mitochondrial architecture are different splice and proteolytic processing forms Mgm1/Opa1, a dynamin-related GTPase at the inner membrane. It is involved in fusion of the mitochondrial inner membrane, and mutations and depletion

lead to severe defects of the structure of cristae (*Sesaki et al., 2003*; *Amutha et al., 2004*; *Frezza et al., 2006*; *Meeusen et al., 2006*). Also proteins involved in lipid transport and metabolism and others whose function is not understood, have been described as affecting mitochondrial architecture (*Messerschmitt et al., 2003*; *Dimmer et al., 2005*; *Connerth et al., 2012*; *Neupert, 2012*; *Tasseva et al., 2013*).

The functional analysis of all these proteins is complicated for a variety of reasons: They can be part of a complex protein interaction network, they can affect protein function through lipid function and vice versa (*Klingenberg, 2009*; *Osman et al., 2011*; *Tasseva et al., 2013*), and some proteins may have dual or multiple different functions which even may influence each other (*Zhang et al., 2003*, *2005*; *Chen et al., 2008*).

We have recently reported a proteomics approach by which mitochondrial proteins can be assigned to different mitochondrial architectural elements (*Harner et al., 2011*). An artificial protein spanning both the outer and inner membrane was introduced that functions as a marker for sites of adhesion. Vesicles were generated by sonic fragmentation of mitochondria and separated by buoyant density centrifugation. We used high resolution mass spectrometry to determine the distribution profiles of proteins across the gradient. In this way, the MICOS complex was identified. In addition, we identified other proteins that meet the criterion of a protein present at contact sites between outer and inner membranes, among them Aim24. Like several subunits of the MICOS complex, Aim24 belongs to a larger group of proteins identified in a screen for altered inheritance of mitochondria (*Hess et al., 2009*). Here we report an analysis of the role of Aim24 in mitochondrial architecture, function and dynamics. Although Aim24 could not be co-isolated with the MICOS complex, our results suggest that Aim24 plays a central role in the modulation of the MICOS complex and in the control of mitochondrial architecture, morphology as well as protein and lipid composition.

## Results

### Aim24 is a protein of the mitochondrial inner membrane

In our previously reported proteomics screen for proteins which accumulate at mitochondrial contact sites we identified a protein exhibiting the same distribution as the subunits of the MICOS complex (*Figure 1A*). The MS/MS data identified it as the product of the gene *AIM24* (systematic name *YJR080C*). This gene had previously been found in a screen for mutants with altered inheritance of mitochondria (*Hess et al., 2009*). It encodes a 391 amino acid protein (44.4 kDa) with unknown function. It was localized to mitochondria in the MitoGFP database (*Huh et al., 2003*) and in a proteomics screen for identification of mitochondrial proteins (*Reinders et al., 2006*). The presence of an N-terminal mitochondrial targeting signal is predicted by the deduced amino acid sequence. To confirm presequence-dependent targeting we analyzed the sizes of wild type Aim24, N-terminally His-tagged Aim24 (inserted after the initial methionine) and C-terminally His-tagged Aim24. In isolated mitochondria N-terminally His-tagged Aim24 showed the same apparent size as wild type Aim24 indicating cleavage of the targeting sequence. In contrast, C-terminally His-tagged Aim24 was slightly larger as it contained the C-terminal His-tag (*Figure 1B*). The processing of the N-terminus of Aim24 strongly suggests import via the TOM-TIM23 pathway and the location of the mature protein in the matrix space or inner membrane. In order to analyse the location of the C-terminus, we treated mitochondria and mitoplasts containing a version of Aim24 carrying a C-terminal HA-tag with proteinase K. Removal of the HA-tag was not observed, suggesting also the presence of the C-terminus in the matrix or in the membrane. Aim24 could only be degraded by proteinase K (*Figure 1C,D*) and several other proteases (not shown) when mitochondria were lysed with Triton X-100.

To test whether Aim24 is firmly associated with the membrane we performed treatment of mitochondria with alkaline solutions of pH up to 12 (*Figure 1D*). The protein was found to be resistant against alkaline extraction. This observation was rather surprising, since the sequence of Aim24 does not contain a predicted transmembrane segment, although a segment lacking charged residues (residues 200–213) is present. To investigate whether Aim24 is anchored in the inner membrane by this segment, we introduced two charged residues (G207R, G208E); according to computational analysis this strongly reduces the hydrophobicity. This mutation did not interfere with targeting of the protein to mitochondria in vivo nor did it alter its lack of extractability by alkaline solutions (*Figure 1E*). Moreover, after alkaline treatment of mitochondria, the mutated Aim24 remained in the membrane and was not present as an aggregate as shown by membrane flotation (*Figure 1F*). Interestingly, in

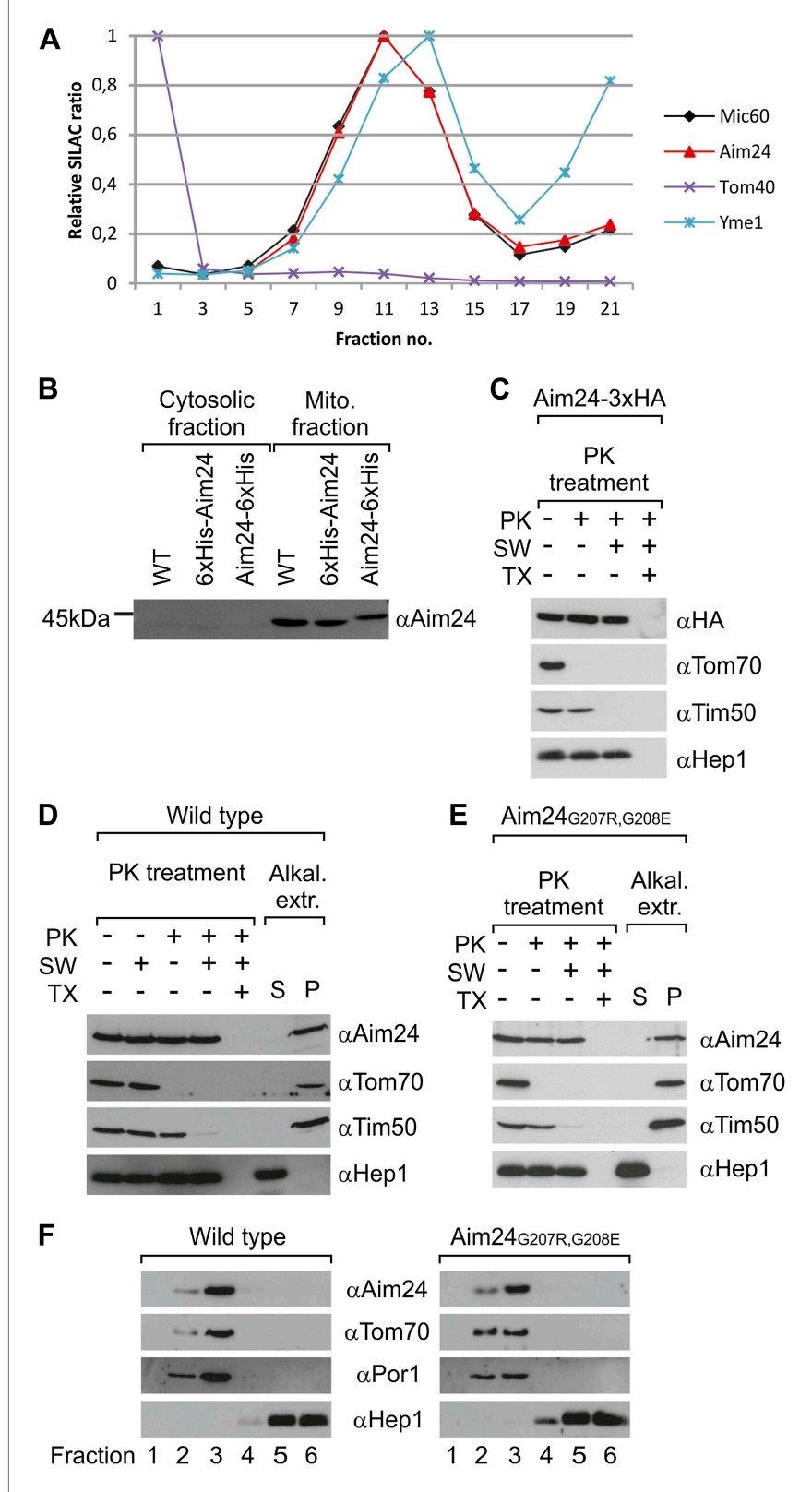

**Figure 1**. Topology of Aim24. (**A**) Aim24 is a contact site protein. Isolated mitochondria from wild type (YPH499) cells were subjected to osmotic shrinking, sonication and buoyant density fractionation. Gradient fractions were analyzed by SILAC supported quantitative mass spectrometry (**Harner et al., 2011**). The distribution of Aim24 in
*Figure 1. Continued on next page*

*Figure 1. Continued*

the gradient is shown in red. Distribution of marker proteins of the OM (Tom40), the IM (Yme1) and of the contact site protein Mic60 are included. (**B**) Aim24 is synthesized as a precursor with an N-terminal targeting signal which is cleaved upon import. Wild type cells harboring pYES2 empty vector, Δ*aim24*harboring pYES2 6xHis-Aim24 or pYES2 Aim24-6xHis were grown on SLac medium containing 0.1% glucose. Protein expression was induced by incubation for 30 min in SLac medium containing 0.1% galactose. Mitochondrial and supernatant fraction were analyzed by SDS-PAGE and immunoblotting with antibodies against Aim24.(**C**) The C-terminal HA-tag of Aim24-3xHA is not accessible to proteinase K (PK) added to intact mitochondria and mitoplasts. Isolated mitochondria were subjected to PK treatment, osmotic swelling (SW) and Triton X-100 (TX) as indicated and analyzed by immunoblotting. (**D**) Aim24 behaves like an integral membrane protein. Left: mitochondria isolated from wild type cells were treated as described in (**C**). Right: mitochondria were exposed to alkaline treatment at pH 12. Soluble (S) and membrane integrated (P, pellet) material was separated by centrifugation and analyzed by immunoblotting. (**E**) Exchange of two glycine residues in a hydrophobic stretch of Aim24 by charged residues does not alter its firm association with the membrane. Mitochondria were isolated from a Δ*aim24* strain harboring the pYES2 plasmid encoding Aim24 G207R, G208E and treated as described in (**C**). (**F**) Neither Aim24 wild type nor Aim24 G207R, G208E protein aggregate upon alkaline treatment of mitochondria. Mitochondria were subjected to alkaline treatment and proteins were separated by flotation gradient centrifugation. The gradients were fractionated (fractions 1–6), proteins were TCA precipitated and analyzed by immunoblotting. Fraction 1 is top and fraction 6 is bottom.

homologs of Aim24 of other species, for example *Kluyveromyces lactis* and *Candida glabrata*, the uncharged segment corresponding to residues 200–213 of *Saccharomyces cerevisiae* Aim24 was even shorter with 12 and 9 residues, respectively. Thus, Aim24 must be integrated in the mitochondrial inner membrane in a different way than by a hydrophobic α-helical transmembrane segment.

## Aim24 is required for the integrity of the MICOS complex

In order to determine whether Aim24 is interacting with the MICOS complex, mitochondria from wild type cells were solubilized with digitonin under conditions that preserve the integrity of the complex and then subjected to size exclusion chromatography. The MICOS complex was eluted early in fractions corresponding to an apparent molecular mass of ca. 1.5 MDa. In addition, minor fractions of Mic10 and Mic26 were found in distinct complexes of smaller size of ca. 250–300 kDa. Aim24 was mainly present in fractions corresponding to 400 kDa (*Figure 2A*), suggesting that it is not an integral or firmly attached part of the MICOS complex.

We tested whether the assembly of the Aim24 complex is affected by deletion of genes encoding MICOS complex subunits. In mitochondria isolated from cells lacking Mic60 or Mic10, the MICOS subunit Mic26 was almost exclusively found in smaller complexes of ca. 250 kDa, suggesting dissociation of the MICOS complex in these strains. In contrast, Aim24 was still present in a complex of the same apparent molecular mass as in wild type (*Figure 2B,C*). Thus, assembly of Aim24 into its complex does not depend on the integrity of the MICOS complex.

Next, we asked whether deletion of the *AIM24* gene affects the integrity of the MICOS complex. Strikingly, with mitochondria from Δ*aim24* cells the majority of Mic10 was found as a ca.300 kDa complex, whereas only a small fraction was present in the 1.5 MDa complex upon gel filtration. The subunits Mic60, Mic27 and the majority of Mic26 were still recovered in the 1.5 MDa complex (*Figure 2D*). This suggests that absence of Aim24 either leads to loss of Mic10 from MICOS, or Mic10 remains with MICOS but is destabilized and Mic10 dissociates upon solubilisation with digitonin. Notably, the expression of C-terminally His-tagged or HA-tagged forms of Aim24 had the same effect on the integrity of the MICOS complex as deletion of the *AIM24* gene; it led to almost complete dissociation of Mic10 upon solubilisation of mitochondria (*Figure 2E*). The MICOS complex may undergo only minor conformational changes due to the minor alterations generated by Aim24 modified by a small C-terminal tag. Therefore, the behavior of Mic10 in Δ*aim24* and Aim24-HA cells appears to represent a destabilization of the MICOS complex. Localization of Mic10 in Δ*aim24* cells by immuno EM analysis is shown in *Figure 2F*. About 93% of all gold particles were located at the IBM, of which 51% at crista junctions. A parallel analysis of wild type mitochondria yielded 95% at the IBM and 62% of these at crista junctions. This does not prove, but is consistent with a continued but weaker association of Mic10 with the MICOS subunits. In conclusion, these results suggest that functional Aim24 is required for the integrity of Mic10 into the MICOS complex.

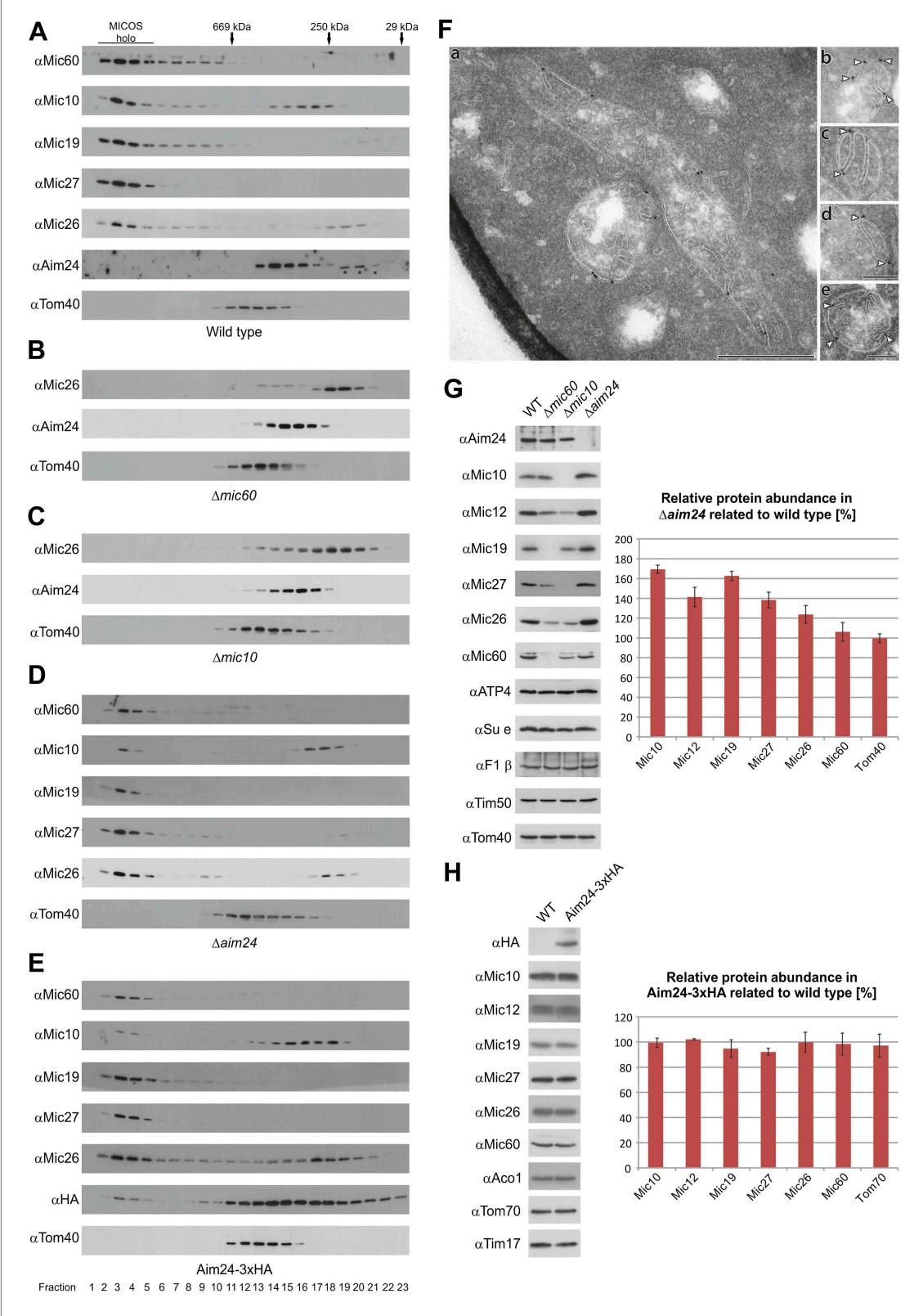

**Figure 2**. Aim24 is required for the integrity of the MICOS complex and for maintenance of steady state levels of MICOS subunits, but Aim24-3xHA is not functional. (**A–E**) Integrity of the MICOS complex was analyzed by molecular sizing chromatography. Mitochondria were isolated from (**A**) wild type, (**B**) Δ*mic60*, (**C**) Δ*mic10*, (**D**) Δ*aim24* and (**E**) Aim24-3xHA cells. They were lysed with digitonin and subjected to gel filtration on a Superose 6 column. *Figure 2. Continued on next page*

*Figure 2. Continued*

Proteins in elution fractions were TCA precipitated and analyzed by SDS-PAGE and immunoblotting using the indicated antibodies. The TOM complex (Tom40) was used as a control. Positions of marker proteins for calibration are indicated by arrows. Asterisks indicate a cross reaction of the Aim24 antibody used. (**F**, a–e) Submitochondrial localization of Mic10 by cryo-immunoelectron microscopy in a strain lacking Aim24 and harboring Mic10-3xHA. Cryo sections were labeled with anti-HA antibodies and proteinA-bound gold particles. Scale bars, 0.2 µm. Arrowheads point at gold particles at crista junctions. (**G**) Steady state levels of MICOS subunits in Δ*aim24* cells. Equal amounts of mitochondrial protein from wild type, Δ*mic60*, Δ*mic10* and Δ*aim24* cells were analyzed by SDS-PAGE and immunoblotting using the indicated antibodies (left panel). Quantitative analysis of protein abundance of MICOS subunits and Tom40 in Δ*aim24* cells in relation to wild type. Error bars indicate standard deviations (right panel). (**H**) Steady state levels of MICOS subunits in Aim24-3xHA. Equal amounts of mitochondrial protein from wild type and Aim24-3xHA cells were analyzed by SDS-PAGE and immunoblotting using the indicated antibodies (left panel). Quantitative analysis of relative protein abundance of MICOS subunits and Tom70 in Aim24-3xHA cells. Error bars indicate standard deviations (right panel).

Do the steady state levels of MICOS components depend on Aim24? In cells in which the *AIM24* gene was deleted, the expression of MICOS subunits was altered. The levels of Mic10, Mic12, Mic27 and Mic26 were increased whereas that of Mic60 was not altered (*Figure 2G*). In contrast, deletion of Mic60 and Mic10 led to down-regulation of the other MICOS subunits. The level of Aim24 in cells lacking Mic60 was unchanged and slightly decreased in cells lacking Mic10. In comparison, the levels of several OM and IM membrane proteins, ATP4, Su e, F1β, Tim50 and Tom40, were not altered (*Figure 2G*). In view of the increased level of Mic10 in Δ*aim24* mitochondria, a fraction of Mic10 in the 300 kDa complex could represent unassembled species rather than protein dissociate upon gel filtration (*Figure 2D*). We quantified the amount of Mic10 present in wild type and Δ*aim24* in the 1.5 MDa (fractions 2–5) and the 300 kDa complex (fractions 16–19) in *Figure 2A,D*. If the amounts of Mic10 present in the 1.5 MDa fractions in wild type and Δ*aim24* are normalized to a ratio of 1, the ratio in the 300 kDa fractions amounts to 5.0. This excludes the possibility that the presence of Mic10 in the 300 kDa fraction was due or solely due to an unassembled form of Mic10. Notably, other MICOS subunits with similar increased levels in Δ*aim24* mitochondria were not present as unassembled species. Finally, we quantified the expression levels of MICOS subunits in Aim24-HA expressing cells. Here, the levels of Mic10 and the other MICOS subunits were not altered as compared to wild type (*Figure 2H*). However, with these mitochondria Mic10 was observed virtually completely in the 300 kDa fraction (*Figure 2E*). This observation strongly supports the notion that the modification of Aim24 by the HA-tag, like the deletion of Aim24, leads to a destabilization of the MICOS complex. In sum, these findings demonstrate the importance of Aim24 for maintaining the integrity of the MICOS complex.

## Aim24 interacts with Mic10

We asked whether Aim24 physically interacts with the MICOS complex. Association was analyzed by co-isolation assays, using digitonin lysed mitochondria containing a His-tagged version of Mic10. While Mic12, Mic19, Mic27, Mic26 and Mic60 could be co-isolated with Mic10, no specific binding of Aim24 was detected (*Figure 3*, left). However, when mitochondria were solubilized with Triton X-100, which leads to dissociation of the MICOS complex into smaller complexes, efficient co-isolation of Aim24 with Mic10 was observed (*Figure 3*, right). This shows that both proteins can specifically interact depending on the state of the MICOS complex. Furthermore, the precise co-fractionation with the MICOS subunits upon fragmentation of mitochondria into small vesicles derived from contact sites is consistent with an association of Aim24 with the MICOS complex (*Figure 1A*). Thus, Aim24 is likely to be weakly or transiently associated with the MICOS complex at mitochondrial contact sites, and Mic10 appears to be involved in this interaction. It should be noted in this context that the MICOS complex itself has a high propensity to fall apart upon solubilisation even with very mild detergents.

## The *AIM24* gene is required for respiratory growth and interacts with genes encoding MICOS subunits

We analyzed the growth of Δ*aim24* deletion mutants on fermentable and non-fermentable carbon sources. On glucose these cells grew like wild type, whereas on non-fermentable carbon sources, such as glycerol and lactate, growth was reduced (*Figure 4A*). Strikingly, cells expressing His-tagged MICOS subunits Mic12 or Mic26 in the Δ*aim24* deletion background did not grow at all on a non-fermentable carbon source (*Figure 4B*). This was surprising since cells containing these His-tagged proteins in the presence of Aim24 displayed no impairment of growth. The combination of two other

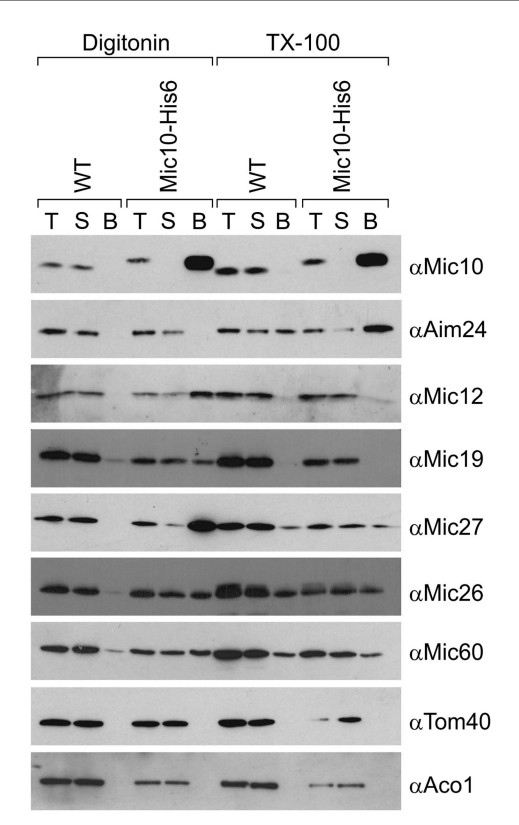

**Figure 3**. Aim24 interacts with Mic10. Mitochondria isolated from wild type and a strain expressing Mic10-His6 were lysed with 1% digitonin or 1% TX-100 and incubated with NiNTA-beads. Bound proteins were eluted using Laemmli buffer containing 300 mM imidazole. Total (T), (5%); supernatant (S), (5%); and bound material (B), (100%) were analyzed by SDS-PAGE and immunoblotting with the indicated antibodies. Aco1, aconitase.

His-tagged MICOS subunits, Mic19-His6 or Mic27-His6, with Δ*aim24* deletion did not result in a synthetic lethal phenotype on non-fermentable carbon source (*Figure 4B*). To relate this to the assembly state of the MICOS complex, we analyzed the His-tagged strains which lacked Aim24 by size exclusion chromatography. In the strains containing Mic12-His6 and Mic26-His6 the MICOS complex was further destabilized, whereas in the strains containing Mic19-His6 and Mic27-His6 the MICOS complex behaved like in the Δ*aim24* mutant containing the wild type forms of the Mic proteins (*Figure 4C*). These results corroborate the conclusion of Aim24 being critical for the assembly of a fully functional MICOS complex. Apparently minor alterations of specific MICOS subunits alter its stability in such a way that in the absence of Aim24 mitochondria are unable to maintain respiratory growth.

## Aim24 is required for assembly of respiratory chain complexes

According to their growth phenotypes cells lacking Aim24 are compromised in their ability to synthesize or maintain the complexes of oxidative phosphorylation. This was not due to the loss of the mitochondrial genome as all strains still contained mtDNA as checked by staining with DAPI (*Figure 5A*). To investigate this phenotype in more detail, we performed Blue native gel electrophoresis to analyse respiratory complexes in isolated mitochondria. Because of the inability of cells harboring His-tagged Mic12 or Mic26 in the Δ*aim24* deletion background to grow on non-fermentable carbon source all strains were grown by adding glucose to lactate containing growth medium. The mitochondria isolated from these cells displayed a drastic reduction or complete absence of the respiratory supercomplexes (*Figure 5B*). There was also a strong reduction of the steady state levels of individual respiratory chain subunits and an almost complete deficiency in subunit e (Su e) of the $F_1F_O$-ATP synthase. Furthermore, the individual MICOS subunits were down-regulated as well when Mic12 or Mic26 were His-tagged in the Δ*aim24* deletion background (*Figure 5C*). Importantly, His-tagging of MICOS subunits or deletion of Aim24 alone did not show a loss of respiratory complexes, nor was this loss observed when His-tagged Mic19 or Mic27 were present in Δ*aim24* mitochondria. This demonstrates the importance of Aim24 for the integrity of the MICOS complex and also the joint requirement of Aim24 and intact MICOS for the formation of the respiratory chain.

## Aim24 controls the composition of cardiolipin and the steady state level of tafazzin

We reasoned that compromised mitochondrial function in Δ*aim24* mutants could be related to an alteration in the homeostasis of mitochondrial membrane lipids. Membrane lipids of mitochondria isolated from Δ*aim24* cells and Δ*aim24* cells carrying His-tagged MICOS subunits were analyzed by mass spectrometry. Mitochondria from wild type, Δ*aim24* as well as Δ*aim24* cells in combination with His-tagged Mic19 or Mic27 did not show a significant difference in the distribution of membrane phospholipids and ergosterol (not shown). In contrast, mitochondria from cells lacking Aim24 and harboring His-tagged Mic12 or Mic26 contained slightly increased levels of ergosterol and, conspicuously,

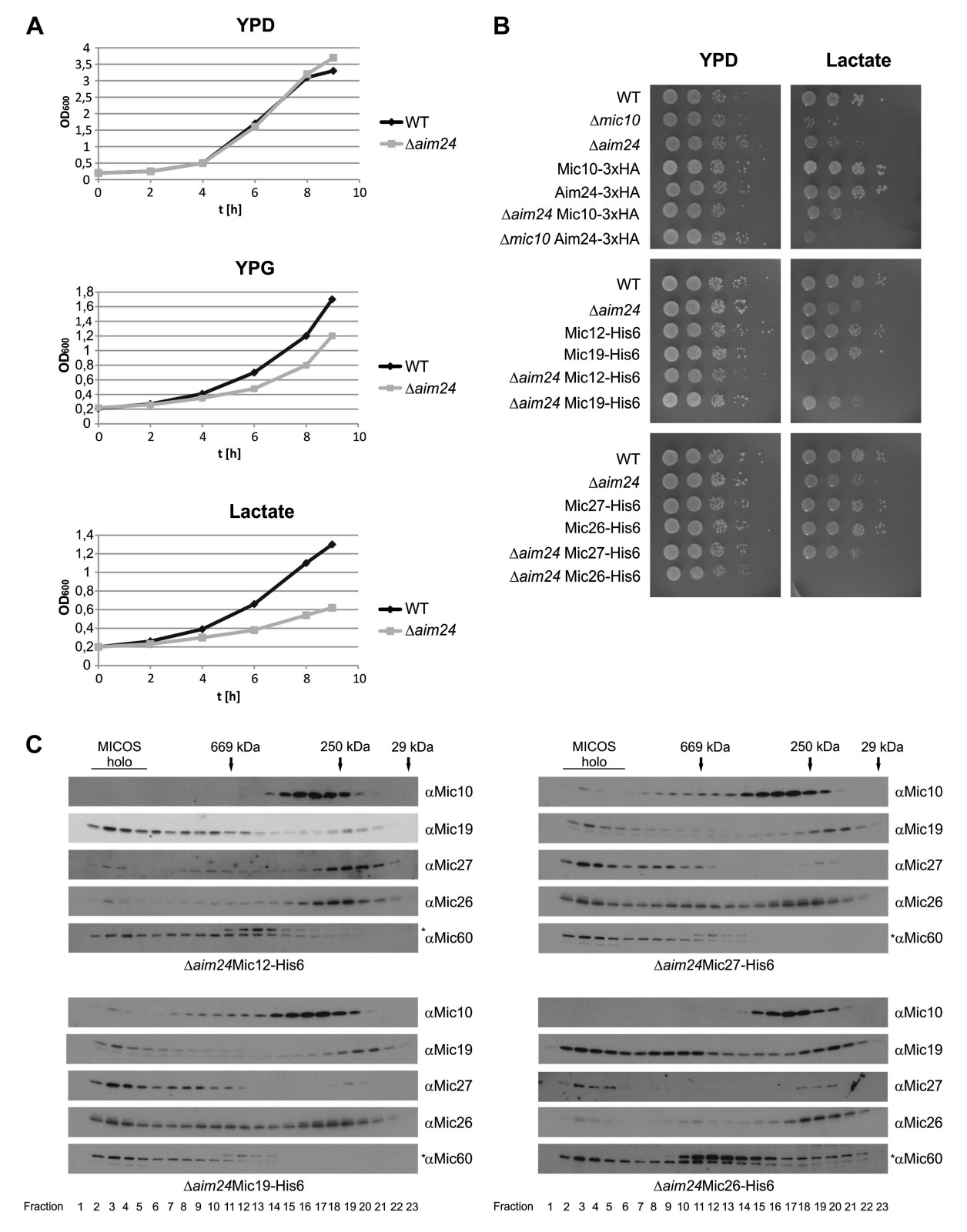

**Figure 4**. Combination of *AIM24* deletion with His-tagging of Mic12 and Mic26, but not of Mic19 and Mic27, leads to respiratory deficiency and instability of the MICOS complex. (**A**) Wild type and Δ*aim24* cells were grown in YPD, YPG or lactate liquid medium at 30°C to logarithmic phase. The cultures were diluted in the respective medium to $OD_{600}$ of 0.2 and further incubated at 30°C. Aliquots were taken every 2 hr and $OD_{600}$ was measured. *Figure 4. Continued on next page*

*Figure 4. Continued*

Black, wild type; grey, Δ*aim24*. (**B**) Growth of the indicated strains on agar plates containing YPD or lactate medium. Cells of the indicated strains were grown in YPD liquid medium at 30°C to logarithmic phase. Serial dilution was performed, 3 µl of each dilution was spotted on the indicated medium and incubated at 30°C. (**C**) Molecular sizing chromatography on Superose 6 column of the indicated strains and immunodetection of the indicated MICOS subunits. Analysis was performed as described in *Figure 2A–E*. Asterisks indicate a cross reaction of the Mic60 antibody used in this analysis.

the pattern of acyl chains specifically of cardiolipin was significantly altered (*Figure 6A*; *Table 1*). There was a shift towards longer and more saturated acyl chains. Since the enzyme responsible for mediating exchange of acyl chains on cardiolipin was identified as tafazzin (Taz1) (*Schlame, 2013*), we checked whether deletion of Aim24 would affect this protein. Mitochondria from wild type and Δ*aim24* cells as well as Δ*aim24* cells carrying His-tagged versions of Mic12, Mic19, Mic27 and Mic26 were analyzed. Notably, the steady state levels of Taz1 were drastically reduced in Δ*aim24* cells expressing Mic12-His6 or Mic26-His6, but not in Δ*aim24* cells expressing Mic19-His6 or Mic27-His6 (*Figure 6B*). Furthermore, cells in which the subunits Mic60 and Mic10 of the MICOS complex were deleted showed wild type levels of Taz1 (*Figure 6C*). Thus, loss of Taz1 is not simply a consequence of the destabilization of the MICOS complex or the loss of crista junctions. These data link together the observed deficiency in respiratory supercomplexes with the function of tafazzin and suggest that the steady state level of tafazzin is coupled to the state of the MICOS complex. Tafazzin was reported to be anchored to the inner face of the outer membrane. Upon gradient fractionation of mitochondrial membrane vesicles it was also found in inner membrane fractions as well as in fractions containing both outer and inner membranes, presumably representing contacts between both membranes (*Ma et al., 2004*; *Brandner et al., 2005*; *Claypool et al., 2006*; *Herndon et al., 2013*). In summary, the MICOS complex and Aim24 may not be directly involved in membrane phospholipid metabolism, but spatially control the process of mitochondrial protein composition which in turn leads to specific changes of the lipid composition.

## Architectural role of Aim24

Since MICOS is a key determinant of mitochondrial inner membrane architecture and MICOS integrity is affected by deletion or mutation of the *AIM24* gene we expected that mitochondrial ultrastructure is altered in the absence of Aim24. The architecture of mitochondria of cells lacking Aim24 was analyzed by electron microscopy.

Upon growth on non-fermentable carbon sources, compared to wild type mitochondria (*Figure 7A, d–f*), mitochondria of Aim24 deficient cells (*Figure 7B, a–i*) showed an increase in the number of inner membrane septae (*Figure 7B, c and f*) (see also *Figure 2F, b–d*). A quantitative evaluation of mitochondrial profiles is presented in *Table 2*. These structures appear like cristae without rims. They likely represent entire or partial separations of the matrix compartment. In a few cases stacks of cristae resembling such as present in Δ*mic60* mitochondria were observed (*Rabl et al., 2009*) (*Figure 7B*, d). A further characteristic was the larger average size of mitochondrial profiles. Particularly impressive was the presence of large mitochondrial structures (*Figure 7B, a, b, g, h, i*; *Table 2*). These mitochondria occupied up to ca. 30–50% of the total cellular area in some images. They contained abundant membranes, which did not have the characteristics of cristae. Instead, they were completely irregular, sheet or tube like. Whereas smaller sized mitochondrial profiles showed regular cristae, in the giant distorted mitochondria their number as well as the length and shape was reduced.

Cells grown on fermentable carbon source showed a drastic increase, as compared to wild type, in the number of septae (*Figure 7A, a–c* and *Figure 7B, j–o*; *Table 2*). No other significant changes in size and shape of mitochondrial profiles were recognized in these cells.

To test whether this phenotype is reflected in the shape of the mitochondrial network we performed fluorescence microscopy using strains expressing GFP targeted to mitochondria. When the Δ*aim24* deletion strain was grown on glucose mitochondrial morphology appeared essentially as in wild type. However, when these cells were grown on glycerol ca. 30% of all cells contained large mitochondrial structures (*Figure 7C*), indicating that Aim24 is required for maintenance of normal mitochondrial morphology under respiratory conditions.

We further analyzed the mitochondria of cells in which the *AIM24* gene was deleted and in addition the MICOS complex destabilized due to the presence of Mic12 or Mic26 as His-tagged versions.

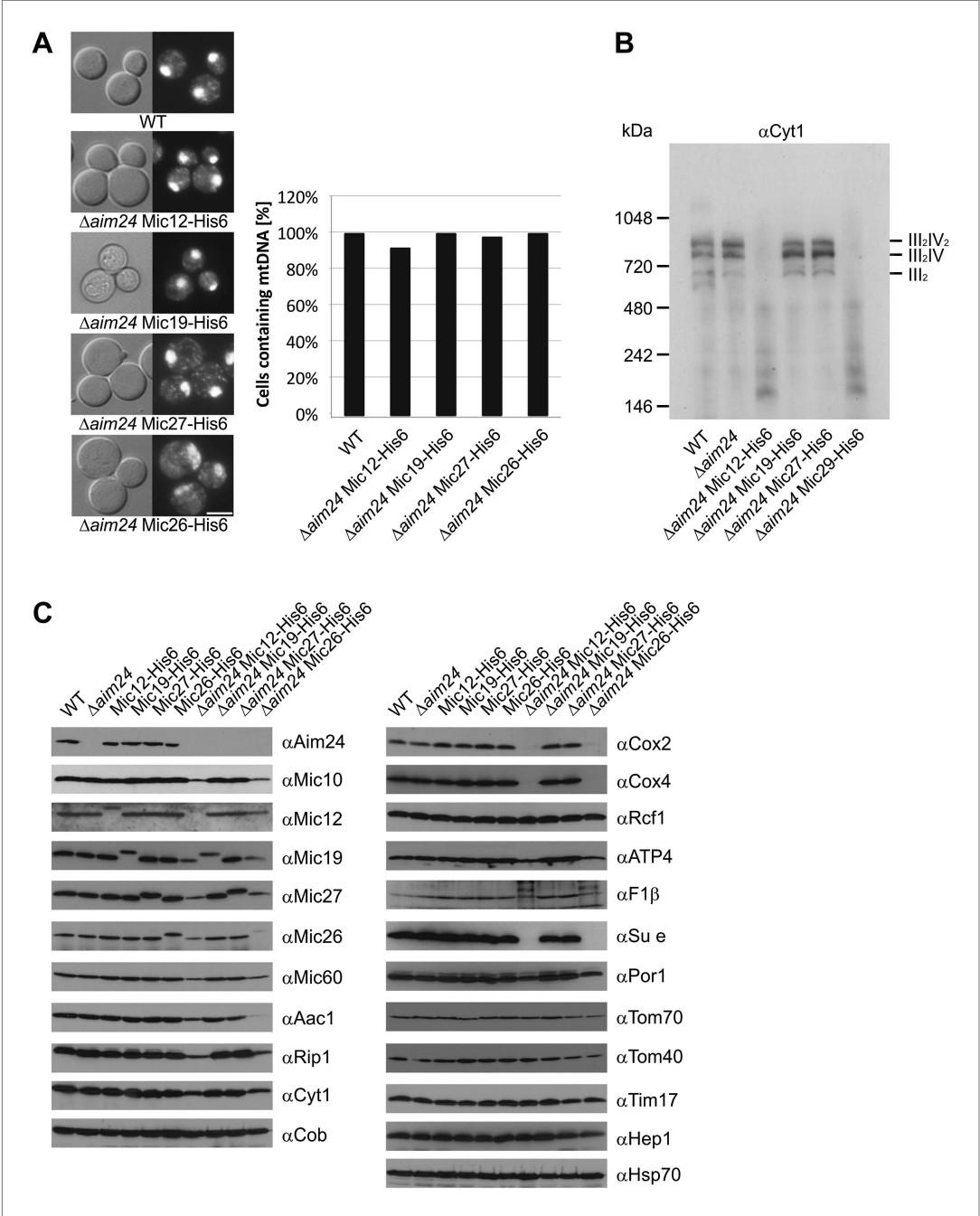

**Figure 5**. Deficiency of respiratory growth of Δ*aim24* mutants are not caused by lack of mitochondrial DNA, but by lack of OXPHOS supercomplexes and of a selective set of proteins involved in ATP production and transport. (**A**) Fluorescence images of DAPI stained cells of the indicated strains. Scale bar, 5 μm (left panel). Quantitative analysis of levels of mitochondrial DNA by DAPI staining (right panel). (**B**) Analysis of respiratory supercomplexes of isolated mitochondria by Blue native gel electrophoresis and immunoblotting with antibodies against cytochrome $c_1$ (Cyt1). The positions of supercomplexes consisting of complex III ($III_2$) and of complexes III plus IV ($III_2IV$ and $III_2IV_2$) are indicated. (**C**) Steady state levels of proteins of mitochondria isolated from the indicated cells grown in the presence of lactate medium containing 0.1% glucose.

These mitochondria displayed virtually complete absence of cristae. They appeared either empty or septae and onion-like structures were present. Mitochondrial profiles were small, and giant mitochondria were completely absent. In contrast, in cells in which MICOS subunits Mic19 or Mic27 were

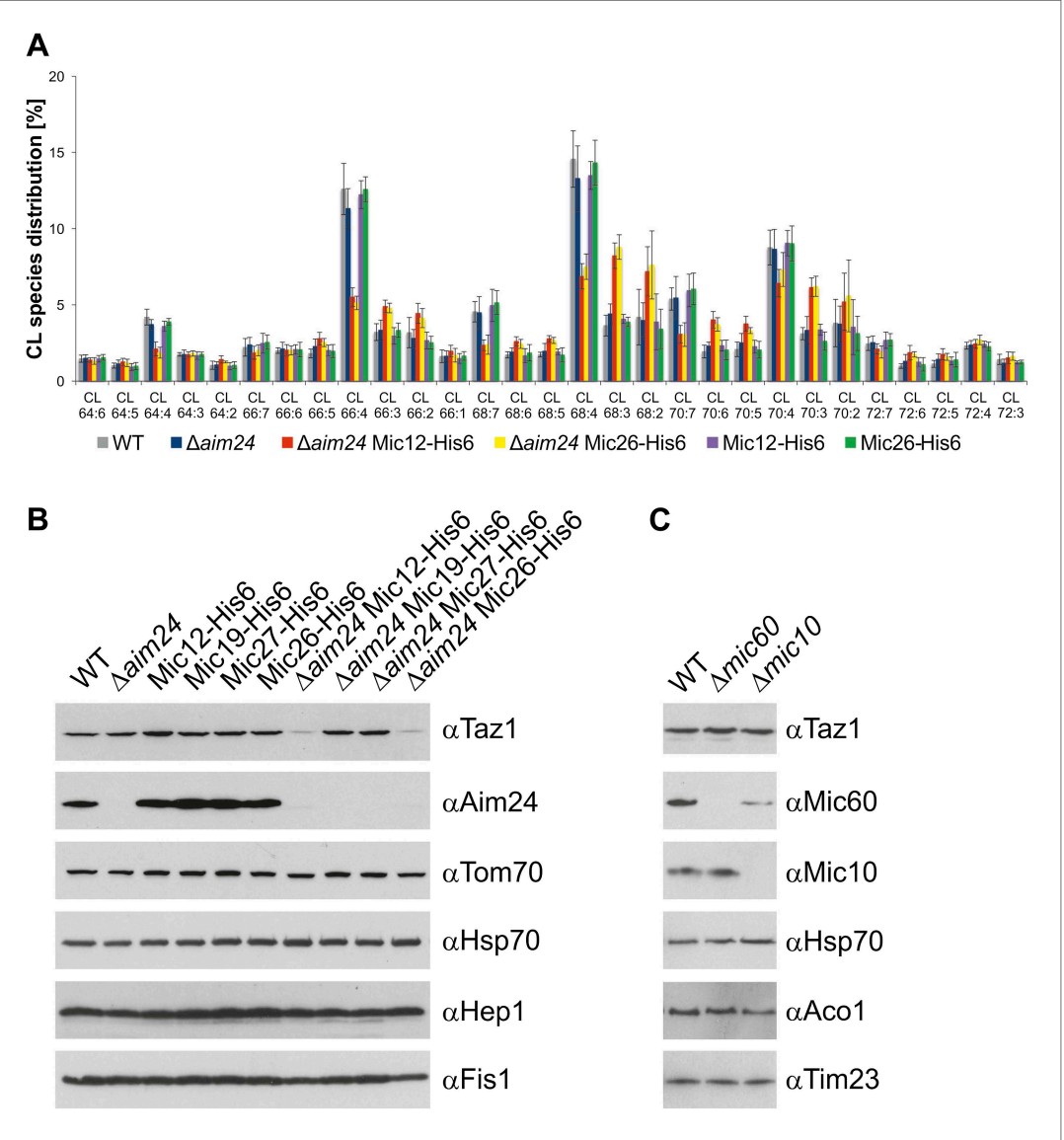

**Figure 6**. Cells lacking Aim24 and containing Mic12-His6 or Mic26-His6 are deficient in remodelling of cardiolipin and are deficient in tafazzin. (**A**) Acyl chain distribution of cardiolipin. First numbers represent the sum of the carbon atoms in the four acyl chains; the numbers behind the colon indicate the total number of double bonds. Abbrevations of strains as in *Figure 5*. (**B**) Tafazzin (Taz1) is strongly down regulated in the absence of Aim24 and presence of His-tagged Mic12 or Mic26. Isolated mitochondria were analyzed by SDS-PAGE and immunodetection of the indicated proteins. (**C**) Presence of Taz1 is not dependent on the presence of Mic60 and Mic10. Isolated mitochondria were analyzed by SDS-PAGE and immunodetection of the indicated proteins.

present in His-tagged form in combination with the Δ*aim24* deletion mitochondria were indistinguishable from mitochondria of cells in which only the *AIM24* gene was deleted (*Figure 8*; *Table 3*). We conclude that Aim24, in an intricate interplay with the MICOS complex, acts as an important modulator in building the architecture and influencing function and integrity of the mitochondria.

## Discussion

In the present study we have combined biochemical, genetic and ultrastructural analysis to characterize a protein of the mitochondrial inner membrane which, according to various criteria, qualifies as an architectural component. Initially detected in a screen for altered inheritance of mitochondria (*Hess et al., 2009*), we found Aim24 in a search for proteins that are present in crista junctions and

**Table 1.** Statistical analysis of the distribution of cardiolipin (CL) species in mitochondria from Δ*aim24* Mic12-His6 and Δ*aim24* Mic26-His6 cells compared to wild type

| CL species | Δ*aim24* Mic12-His6 | Δ*aim24* Mic26-His6 |
|---|---|---|
| 64:6 | ns | ns |
| 64:5 | ns | ns |
| 64:4 | * 0.015 | ** 0.0037 |
| 64:3 | ns | ns |
| 64:2 | * 0.0416 | ns |
| 66:7 | ns | ns |
| 66:6 | ns | ns |
| 66:5 | ** 0.0014 | ** 0.0030 |
| 66:4 | ** 0.0023 | ** 0.0014 |
| 66:3 | ** 0.0038 | ** 0.0047 |
| 66:2 | ns | ns |
| 68:7 | *** <0.0001 | *** 0.0002 |
| 68:6 | *** 0.0001 | *** 0.0006 |
| 68:5 | *** <0.0001 | *** <0.0001 |
| 68:4 | *** <0.0001 | *** <0.0001 |
| 68:3 | *** <0.0001 | *** <0.0001 |
| 68:2 | * 0.0130 | * 0.0156 |
| 70:7 | *** 0.0001 | *** 0.0003 |
| 70:6 | *** <0.0001 | *** <0.0001 |
| 70:5 | *** <0.0001 | *** 0.0002 |
| 70:4 | ** 0.0028 | ns |
| 70:3 | *** <0.0001 | *** <0.0001 |
| 70:2 | ns | ns |
| 72:7 | ns | * 0.0480 |
| 72:6 | ** 0.0013 | *** <0.0001 |
| 72:5 | ** 0.0055 | ** 0.0057 |
| 72:4 | ns | * 0.0359 |
| 72:3 | ns | ns |

Mitochondria from cells of the respective strains grown on lactate medium containing 0.1% glucose were isolated. Lipids were extracted and analyzed by mass spectrometry. Analyses contain three biological and technical replicates. The first numbers of CL species represent the sum of the carbon atoms in the four acyl chains; the numbers behind the colon indicate the total number of double bonds. Statistics were calculated by an unpaired *t* test using GraphPadPrism; two-tailed p-values with a 95% confidence interval are given with *$p<0.05$, **$p<0.01$, ***$p<0.001$.

associated with contact sites that link outer and inner boundary membranes (*Harner et al., 2011*). We report here that Aim24 has a distinct function in stabilizing/modulating the MICOS complex which is relevant in a number of aspects such as protein and lipid composition, architecture and metabolic function of mitochondria. Aim24 is firmly associated with the mitochondrial inner membrane. This was unexpected in view of the overall hydrophilic nature of the protein and the absence of a predicted transmembrane segment. Computational structure prediction of Aim24 indicates that it belongs to the family of TRAP-like proteins. A hallmark of this family is that its members are composed of antiparallel β-sheets (*Antson et al., 1995*). So far no protein with a domain consisting of β-sheets has been described to exist in the inner membrane of mitochondria; however, it cannot be excluded that Aim24 is embedded in the lipid environment by this structural element. This would explain the firm association with the inner membrane. At the same time, this might explain the existence of Aim24 as a large complex of ca. 400 kDa as the members of the TRAP-like protein family were reported to form oligomers.

The important role of Aim24 is due to its intimate relationship to the MICOS complex. In the absence of functional Aim24, the MICOS complex is destabilized as demonstrated by the dissociation of Mic10 upon detergent solubilisation of mitochondria. Interestingly, a physical interaction of Aim24 with subunits of the wild type MICOS complex was not observed after lysis of mitochondria with digitonin. However, when the MICOS complex was dissociated by Triton X-100, Aim24 was found to be completely associated with Mic10. In the intact mitochondria Aim24 is apparently associated with the MICOS complex, in consistence with its topology indicated by our quantitative proteomics screen (*Harner et al., 2011*). This interaction is obviously highly sensitive to minor disturbances, such as lysis of mitochondrial membranes even with mild detergents and the presence of C-terminal tags on certain MICOS subunits. The interaction is also reflected in the strong alteration of the expression levels of the MICOS subunits in cells deficient in Aim24, most of them being up-regulated. Thus, Aim24 is likely to regulate the association of Mic10 with other subunits of the complex. One would, however, not exclude the possibility that Aim24 exerts its function also as an assembly factor of the MICOS complex.

Deletion of Aim24 leads to distinct functional deficiencies of the mitochondria. In agreement with the reported alteration of inheritance of mitochondria (*Hess et al., 2009*), Aim24 deficient cells are restricted in their growth on non-fermentable carbon source. Interestingly, combination of deletion of Aim24 with the expression of each of the two MICOS subunits Mic12 or Mic26 as C-terminally His-tagged variants leads to

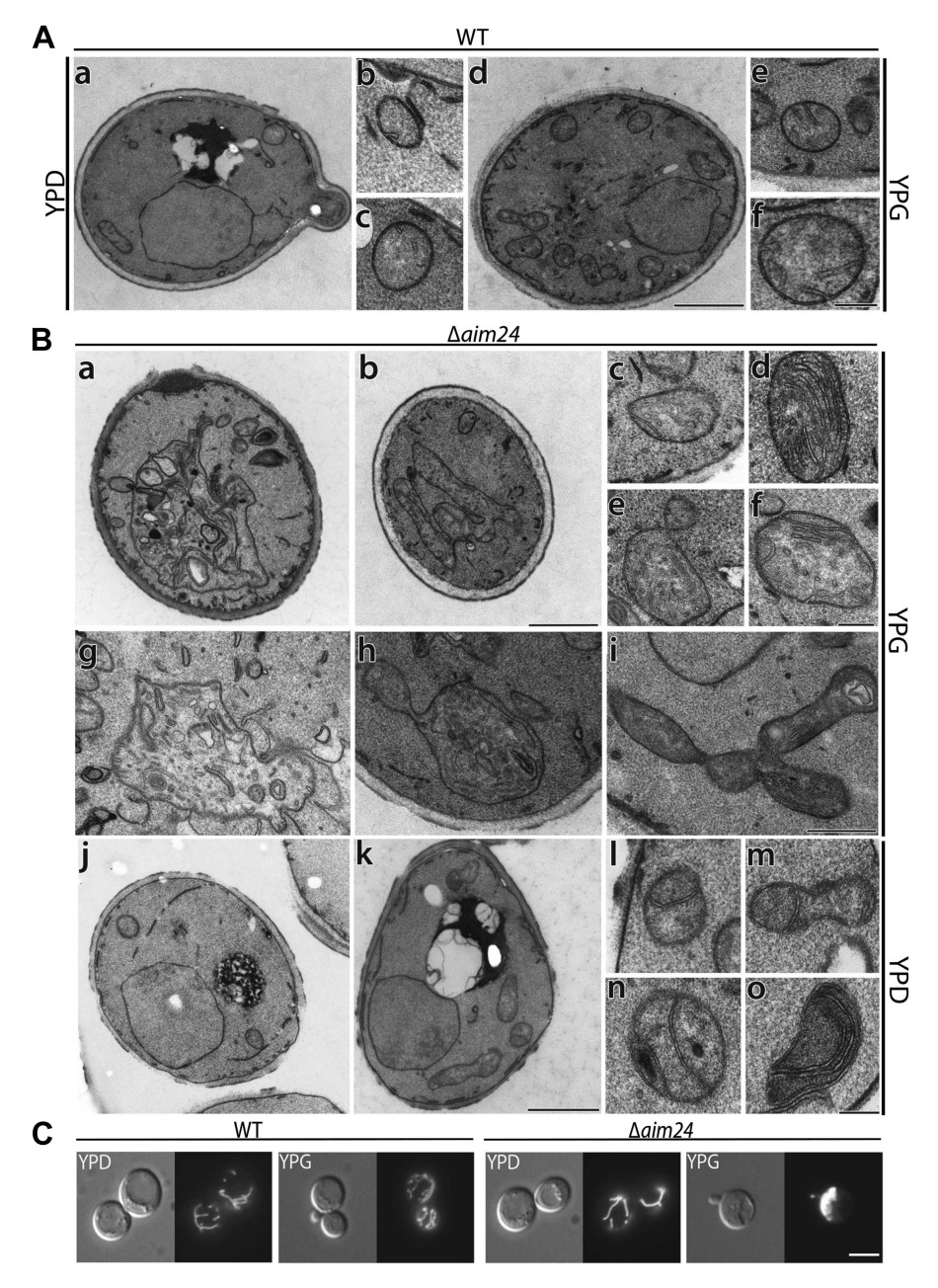

**Figure 7**. Deletion of the *AIM24* gene leads to altered mitochondrial morphology and architecture. (**A** and **B**) Electron microscopy of wild type and *AIM24* deletion mutant. Cells were grown in either YPD or YPG and fixed with glutaraldehyde, contrasted with osmium tetroxide and uranylacetate. Ultrathin sections were obtained and contrasted with uranylacetate and lead citrate. (**A**) Wild type. a–c, cells grown on YPD; d–f cells grown on YPG; a and d, cells (scale bar, 1 μm); b, c, e, f, mitochondrial profiles (scale bar, 0.2 μm). (**B**) Δ*aim24*. a–i, cells grown on YPG; j–o cells grown on YPD; a, b, j and k, cells (scale bar, 1 μm). c–i and l–o, mitochondrial profiles (scale bars, 0.2 μm). (**C**) Fluorescence microscopy of wild type and Δ*aim24* cells expressing mitoGFP. Cells were grown at 30°C on YPD or YPG medium to stationary phase (scale bar, 5 μm).

synthetic lethality on non-fermentable carbon sources. This points to a functional interaction of these genes as the single deletion mutants are either not at all or only mildly affected (*Harner et al., 2011*). This supports the conclusion that Aim24 has a very important role in controlling the function of the MICOS complex.

**Table 2.** Quantitative analysis of ultrastructural parameters of wild type cells (strain YPH499) and cells in which *AIM24* was deleted

| Strains and media | WT, YPD | WT, YPG | Δ*aim24*, YPD | Δ*aim24*, YPG |
|---|---|---|---|---|
| Mitochondrial profiles | 115 | 302 | 126 | 170 |
| Profiles with a diameter >0,66 μm [%] | 8,7 | 10,3 | 19,0 | 34,7 |
| Profiles with a diameter >1,33 μm [%] | 1,7 | 1,0 | 4,0 | 15,9 |
| Septae [%] | 4,3 | 3,6 | 24,6 | 7,6 |

Cells were grown on glucose (YPD) or glycerol (YPG) containing media. 50 cells were randomly selected, the number and diameter of mitochondrial profiles determined as well as the number of septae. The values were normalized to 100 mitochondrial profiles (%).

Aim24 is a protein with a distinct architectural role. This is not only suggested by its interaction with the MICOS complex, but also by prominent phenotypical traits that become apparent upon deletion or modification of Aim24. First, the percentage of mitochondria with septae is strongly increased, as seen in cells grown on fermentable and non-fermentable carbon source. And second, giant distorted mitochondria accumulate with irregular inner membranes. In particular the formation of inner membrane septae indicates a role of Aim24 in the shaping of crista junctions. These are usually narrow ring- or slot-like structures (*Mannella et al., 2001*; *Rabl et al., 2009*). Their extension leads to structures which in sections show double membranes crossing the matrix completely, and which appear as septae. Notably, the width of the junctions of cristae and septae at the inner boundary membrane were like in wild type, whereas their lateral extension was strongly increased. The formation of giant mitochondria is apparent both in fluorescent and electron microscopic images. These mitochondria show a dramatically altered structure of their inner membrane. We consider it likely that they represent later stages of inner membrane deterioration concomitant with accumulation of inner membrane defects.

The effect of tagging specifically the MICOS subunits Mic12 or Mic26 in the Δ*aim24* deletion background is dramatic. Introducing one of these altered subunits into Δ*aim24* cells leads to the absence of the large and giant mitochondria typical of Δ*aim24* cells and presence of particularly small mitochondria, entirely lacking cristae and instead containing septae. This is accompanied by the loss of complexes and components of oxidative phosphorylation and the inviability under respiratory conditions. Although the over all levels of cardiolipin and of other membrane phospholipids is not altered in these latter mutants, the difference in the pattern of acyl chains in cardiolipin is remarkable. The massive reduction of respiratory complexes whose function is dependent on cardiolipin indicates a connection between Aim24, cardiolipin and the function of the MICOS complex. Surprisingly, the alteration in the types of acyl residues observed were characteristic of those described for mutants with deficiencies in tafazzin in different organisms (*Vreken et al., 2000*; *Schlame et al., 2002*; *Gu et al., 2004*). Deficiency in tafazzin underlies the Barth syndrome, a severe X-chromosome-linked disorder characterized by cardiac and skeletal myopathy and neutropenia (*Barth et al., 1983*). In mitochondria of patient cells the fraction of cardiolipin with tetralinoleoyl groups is largely diminished. Furthermore, alterations of the structure of cristae and partial deficiency of respiratory complexes were observed in humans and also in several different organisms including yeast (*Joshi et al., 2009*; *Acehan et al., 2011*). Tafazzin is a cardiolipin transacylase (*Xu et al., 2006*; *Schlame, 2013*). It was suggested to be recruited to non-bilayer type lipid domains that occur in curved membrane areas sensing non-bilayer arrangement of membrane lipids (*Schlame et al., 2012*). Mitochondrial cristae junctions and contact sites, structured by the MICOS complex correspond exactly to such proposed areas. Interestingly, the human homolog of Mic26, APOOL, has recently been shown to have the ability to bind cardiolipin (*Weber et al., 2013*).

We report here that mitochondria lacking Aim24 and containing either His-tagged Mic12 or Mic26 not only show the characteristic alterations of cardiolipin resulting from tafazzin depletion, but in addition almost completely lack tafazzin. Surprisingly, beside the drastic changes in mitochondrial architecture, there is a practically complete loss of respiratory complexes. This phenotype is much more severe than that reported for tafazzin deficiency, but can be explained by the complete loss of cristae in these mutant cells. Putting all the available information together we propose that the

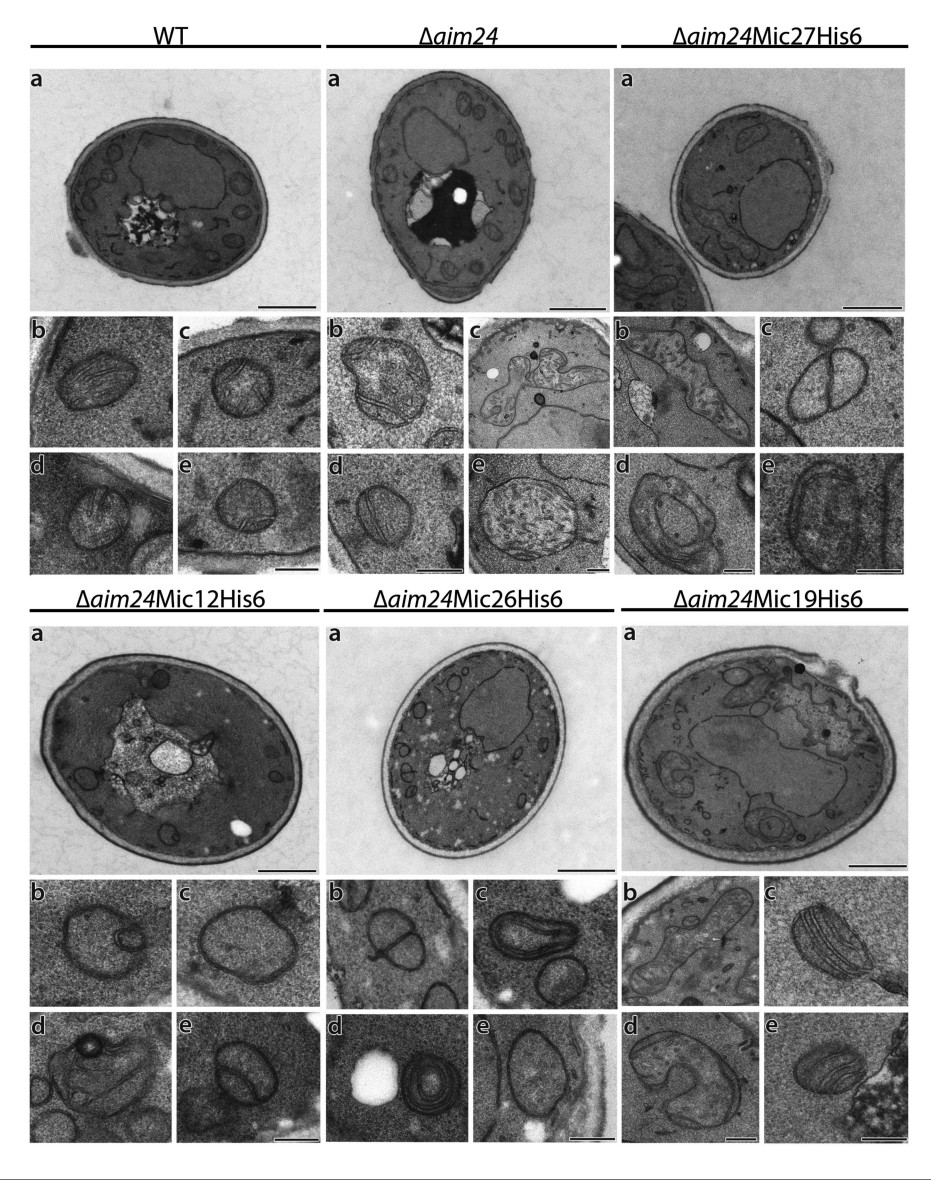

**Figure 8**. His-tagging of Mic12 and Mic26, but not of Mic19 and Mic27, in Δ*aim24* background leads to mitochondria grossly deficient in cristae and accumulation of septae and onion-like structures. Cells were grown on YPG plus 0.1% glucose and electron microscopy was performed as in *Figure 7A,B*. a, cells (scale bars, 1 μm); b–e, mitochondrial profiles (scale bars, 0.2 μm).

localization of tafazzin in the outer membrane provides it with the ability to reach the inner membrane specifically at contact sites and crista junctions. On the one hand, this would explain the partial co-localization with inner membrane fractions (*Claypool et al., 2006*; *Herndon et al., 2013*). On the other hand in this way tafazzin could become enzymatically active directly at crista junctions, membrane sites showing extreme curvature where cardiolipin modified by tafazzin is specifically needed (*Schlame et al., 2012*). Such a mechanism may be relevant also in regard to the important question raised previously, why and how only a specific limited fraction of cardiolipin is modified by tafazzin (*Schlame, 2013*). In conclusion, Aim24 appears to have an important role in determining mitochondrial architecture by affecting not only assembly of the MICOS complex, but also by organizing complex reactions such as the remodelling of cardiolipin which makes it competent for the formation of respiratory supercomplexes and the architecture of cristae.

**Table 3.** Quantitative analysis of ultrastructural parameters of cells deficient in *Aim24* and carrying one of the MICOS subunits Mic12, Mic19, Mic27 or Mic26 in His-tagged form

| Strains | WT | Δaim24 | Δaim24 Mic12-His6 | Δaim24 Mic19-His6 | Δaim24 Mic27-His6 | Δaim24 Mic26-His6 |
|---|---|---|---|---|---|---|
| Mitochondrial profiles | 246 | 182 | 133 | 214 | 202 | 184 |
| Profiles with a diameter >0,66 μm [%] | 5,7 | 13,7 | 1,5 | 10,8 | 7,9 | 2,7 |
| Profiles with a diameter >1,33 μm [%] | 0,8 | 5 | 0,0 | 3,3 | 1,5 | 1,6 |
| Septae [%] | 1,2 | 14,3 | 22,6 | 16,8 | 15,8 | 26,1 |
| Onion like [%] | 0 | 0,6 | 7,5 | 0,5 | 0,5 | 7,6 |
| Without cristae [%] | 14,2 | 9,9 | 61,7 | 7,9 | 6,4 | 53,8 |
| Crista junctions [%] | 68,7 | 65,4 | 3,8 | 49,5 | 53,5 | 2,2 |

Cells were grown on YPG supplemented with 0.1% glucose to allow comparison of the cells with His-tagged Mic12 and Mic26 which are unable to grow on YPG. 50 cells of each strain were randomly selected. The number of mitochondrial profiles, the diameter of mitochondrial profiles, the number of septae, onion-like structures, mitochondria without internal membranes and the number of crista junctions were determined. The values were normalized to 100 mitochondrial profiles (%).

Finally, our study illustrates the difficulties in the attempt to determine the relevance of molecular architecture of organelle function. This is obviously due to a number of factors, such as the multiplicity of diverse functions of individual proteins, the intricate and flexible interactions of proteins and membrane lipids, compensatory genetic effects as well as complex mechanisms of organellostasis. Future research on the molecular architecture of mitochondria will have to integrate these various determinants of this complex regulatory network.

# Materials and methods

## Yeast strains and cell growth

*S.cerevisiae* strain YPH499 {ura3-52, lys2-801[amber], ade2-101[ocre], trp1-Δ63, his3-Δ200, leu2-Δ1} was used as the wild type (WT). All chromosomal manipulations (knock outs, C-terminal 6xHis- and 3xHA-tagging) were performed according to established procedures (*Longtine et al., 1998*; *Knop et al., 1999*). The coding region of Aim24 was replaced using the pFA6a-His3MX6 plasmid as PCR template. The 6xHis-tag and the 3xHA-tag were introduced using the pYM9 or pYM2 plasmid as the PCR template. Double mutant strains were also generated by homologous recombination. The genotypes of the various strains are listed in *Table 4*.

For generation of pYES2Aim24 constructs the coding region of Aim24 was amplified by PCR. The N- and C-terminal 6xHis tags were inserted into pYES2 using the *Eco*RI and *Not*I restriction sites. The point mutations G207R G208E in Aim24 were introduced using the QuickChange Site-Directed Mutagenesis Kit (Stratagene, La Jolla, CA, USA) according the manufacturer's instructions.

Growth of WT and YPH499Δaim24 was analyzed in liquid culture. Cells were cultured at 30°C in YPD (1% yeast extract, 2% peptone, 2% glucose), YPG (1% yeast extract, 2% peptone, 3% glycerol) or lactate liquid medium (Lac) (3 g yeast extract, 1 g $NH_4Cl$, 1 g $KH_2PO_4$, 0.5 g $CaCl_2*2H_2O$, 0.5 g NaCl, 1 g $MgSO_4*7H_2O$, and 3 mg $FeCl_3$ per liter) (*Sherman, 1991*). Cultures were kept in the logarithmic growth phase except otherwise indicated. For growth analysis the cultures were diluted to an $OD_{600}$ of 0.2 and the $OD_{600}$ was recorded at the indicated time points. For all further growth analysis drop dilution assays were performed. Cells were cultured in YPD liquid medium until they reached the logarithmic (log) phase, washed with water, diluted in water to an $OD_{600}$ of 0.3 and serial dilution was performed (1:10; 1:100; 1:1000; 1:10,000). 3 μl of each dilution were spotted on agar plates containing either YPD medium or lactate medium and were incubated at 30°C.

## Proteomics data analysis

High resolution Orbitrap data (*Harner et al., 2011*) were analyzed in the Max Quant software environment (*Cox and Mann, 2008*) using protein correlation profiling (*Andersen et al., 2003*).

**Table 4.** *S. cerevisiae* mutant strains generated for this study

| Strain | Genotype |
| --- | --- |
| Δ*aim24* | MATα, ura3-52, lys2-801^amber, ade2-101^ocre, trp1-Δ63, his3-Δ200, leu2-Δ1, aim24::HIS3 |
| Aim24-His6 | MATα, ura3-52, lys2-801^amber, ade2-101^ocre, trp1-Δ63, his3-Δ200, leu2-Δ1, Aim24-His6-KAN |
| Aim24-3xHA | MATα, ura3-52, lys2-801^amber, ade2-101^ocre, trp1-Δ63, his3-Δ200, leu2-Δ1, Aim24-3xHA- HIS3 |
| Δ*aim24* Mic10-3xHA | MATα ura3-52, lys2-801^amber, ade2-101^ocre, trp1-Δ63, his3-Δ200, leu2-Δ1, aim24::HIS3, Mic10-3xHA-KAN |
| Δ*aim24* Mic12-His6 | MATα ura3-52, lys2-801^amber, ade2-101^ocre, trp1-Δ63, his3-Δ200, leu2-Δ1, aim24::HIS3, Mic12-His6-KAN |
| Δ*aim24* Mic19-His6 | MATα ura3-52, lys2-801^amber, ade2-101^ocre, trp1-Δ63, his3-Δ200, leu2-Δ1, aim24::HIS3, Mic19-His6-KAN |
| Δ*aim24* Mic27-His6 | MATα ura3-52, lys2-801^amber, ade2-101^ocre, trp1-Δ63, his3-Δ200, leu2-Δ1, aim24::HIS3, Mic27-His6-KAN |
| Δ*aim24* Mic26-His6 | MATα ura3-52, lys2-801^amber, ade2-101^ocre, trp1-Δ63, his3-Δ200, leu2-Δ1, aim24::HIS3, Mic26-His6-KAN |

All chromosomal manipulations were performed by homologous recombination using the indicated marker cassettes. Left column, names of the various strains; right column, genotypes of the respective strains.

## Generation of antibodies

The protein segments 16–119 of *S.cerevisiae* Aim24 or 1–117 of *S.cerevisiae* Taz1 were expressed in *E.coli* using the pQE vector system (Qiagen, Hilden, Germany). The proteins were purified by Ni-NTA chromatography and injected into rabbits. The generated antibodies were purified as described (*Harlow and Lane, 1988*).

## Isolation of mitochondria

Cells were grown on lactate medium with or without 0.1% glucose as indicated at 30°C. Mitochondria were isolated by differential centrifugation as described previously (*Herrmann et al., 1994*).

## Fluorescence microscopy

Mitochondria in living yeast cells were stained with mitochondria-targeted GFP expressed from pVT100U-mtGFP (*Westermann and Neupert, 2000*). Yeast cells were grown to stationary phase in YPD or YPG medium at 30°C. For quantification 100 cells of each strain were selected randomly. Microscopy was performed with an Axioplan 2 wide field fluorescence microscope (Carl Zeiss Lichtmikroskopie, Göttingen, Germany) equipped with an Evolution VF Mono Cooled monochrome camera (Intas, Göttingen, Germany) with Image ProPlus 5.0 and Scope Pro 4.5 software (Media Cybernetics, SilverSpring, MD, USA). Image adjustment of brightness and contrast was performed using Adobe Photoshop 13.0 (Adobe Systems Software, San Jose, CA, USA).

## DAPI staining

Cells were grown to log phase in YPD medium, harvested and fixed in methanol for 5 min at room temperature. For DNA staining cells were washed with PBS, incubated with 1 µg/µl diaminophenylindole (DAPI) in PBS for 5 min at room temperature, rinsed four times with PBS and finally resuspended in 350 µl PBS. For quantification 100 cells of each strain were selected randomly. Micrographs were taken using an Axiophot microscope (Carl Zeiss Lichtmikroskopie, Göttingen, Germany) equipped with a Leica DCF360FX Camera with Leica LAF AF Version 2.2.1 Software (Leica Microsystems, Wetzlar, Germany).

## Standard electron microscopy

Cells were grown to log phase as indicated in either YPG medium or YPG medium containing 0.1% glucose and prepared for electron microscopy as described previously (*Bauer et al., 2001*). Ultrathin sections of 50–70 nm were obtained using an Ultracut EM UC6 ultramicrotome (Leica Microsystems, Wetzlar, Germany) with a diamond knife (type Ultra 45°; Diatome, Biel, Switzerland), collected on Pioloform-coated copper slot grids (Plano, Wetzlar, Germany) and stained with uranyl acetate and lead citrate (*Reynolds, 1963*). Samples were analyzed using an EM 902A (Carl Zeiss, Oberkochen, Germany) transmission electron microscope operated at 80 kV. Micrographs were taken using a 1.350 × 1.050 pixel Erlangenshen ES500W CCD camera (Gatan, Pleasanton, CA, USA) and Digital Micrograph software (version 1.70.16, Gatan).

## Cryo-immuno electron microscopy

Cells were grown to exponential phase in lactate medium, chemically fixed, embedded in 12% gelatin and cryo-sectioned as described previously (*Griffith et al., 2008*). Ultrathin sections of 50–70 nm were cut using an Ultracut EM UC6 with EM FC6 cryochamber (Leica Microsystems, Wetzlar, Germany) and cryo-immuno diamond knife (Diatome, Biel, Switzerland). Sections were collected with a 1:1 mixture of 2% methylcellulose (in ddH$_2$O) and 2.3 M sucrose (in PHEM buffer) using Formvar/carbon coated copper 100 mesh grids.

For immune EM, cryo-sections were post labeled using monoclonal anti-HA (1:60 in blocking buffer containing 1% BSA, 0.5% CWFSG and 0.01% BSA-C in PBS) and rabbit anti mouse antibodies (1: 250 in blocking buffer) and a protein A-gold 10 nm conjugate (in 1% BSA in PBS) before being viewed in a Jeol 1010 electron microscope (Jeol, Tokyo, Japan) operated at 80 kV. Micrographs were taken using Kodak 4489 sheet film (Eastman Kodak, Rochester, NY, USA).

## Proteolysis protection assay

Mitochondria (50 µg protein) were incubated in 45 µl of either SM buffer (0.6 M sorbitol, 20 mM MOPS, pH 7.4), swelling buffer (20 mM MOPS, pH 7.4) or lysis buffer (1% Triton X-100, 20 mM MOPS, pH 7.4) in presence or absence of proteinase K (final concentration of 0.2 mg/ml) for 30 min on ice. Proteinase K was inactivated by the addition of phenylmethanesulfonyl fluoride (PMSF) to a final concentration of 4 mM and incubation for 10 min on ice. Samples were spun down for 20 min (20,000 × g, 4°C), resuspended in SM buffer and precipitated by addition of trichloroacetic acid (TCA) at a final concentration of 14%. The precipitates were resuspended in Laemmli buffer, subjected to SDS-PAGE and analyzed by immunoblotting.

## Alkaline extraction

Mitochondria (100 µg protein) were resuspended in 75 µl SM buffer and 75 µl 200 mM Na$_2$CO$_3$ were added. After incubation for 30 min on ice, samples were centrifuged for 30 min (135,000×g, 4°C). The pellets were resuspended in Laemmli buffer, 30% of the supernatants were subjected to TCA precipitation and resuspended in Laemmli buffer. Samples were analyzed by SDS-PAGE and immunoblotting. For membrane flotation the mitochondria were treated with Na$_2$CO$_3$ and placed under two layers of buffer containing 0.25 M and 1.4 M sucrose in 20 mM MOPS, pH7.4, and centrifuged for 3 hr at 120,000×g at 4°C. The gradients were fractionated and proteins precipitated with TCA. Proteins were analyzed by SDS-PAGE and immunoblotting.

## Size exclusion chromatography

Mitochondria (1 mg protein) were suspended in 500 µl digitonin containing buffer (20 mM Tris–HCl, pH 7.4, 50 mM NaCl, 0.1 mM EDTA, 10% [vol/vol] glycerol, 1 mM PMSF, 1% digitonin). After solubilisation for 30 min on ice a clarifying spin was performed (10 min at 12,000 × g, 4°C). Cleared lysates were subjected on Superose 6 size exclusion column (GE Healthcare, Piscataway, NJ, USA; elution buffer 20 mM Tris, pH 7.4, 150 mM NaCl, 5% [vol/vol] glycerol and 0.075% [wt/vol] digitonin). Fractions were analyzed by SDS-PAGE and immunoblotting. To calibrate the column, a mixture of thyroglobulin, catalase and carboanhydrase was subjected to the same procedure.

## Co-isolation (pull-down) assay

Mitochondria (1 mg protein) isolated from WT and a strain expressing Mic10-His6 were lysed in 1 ml lysis buffer (50 mM potassium phosphate, 50 mM NaCl, 10 mM imidazole, pH 8.0, 1 mM PMSF) including either 1% (wt/vol) digitonin or 1% (vol/vol) Triton X-100. Ni-NTA affinity chromatography was performed and fractions were analyzed by SDS-PAGE and immunoblotting.

## Analysis of steady state levels of mitochondrial proteins

For analysis of the steady state levels of different mitochondrial proteins equal amounts of mitochondria of the respective strains were subjected to SDS-PAGE and analyzed by immunoblotting.

## Blue native gel electrophoresis (BN-PAGE)

Mitochondria (150 µg protein) were pelleted by centrifugation (12 min at 17,600×g, 4°C) and resuspended in 15 µl BN-lysis buffer (50 mM NaCl, 50 mM imidazole, 2 mM 6-aminohexanoic acid, 1 mM EDTA, 1 mM PMSF, 3% digitonin, pH 7.0) (*Wittig et al., 2006*). After incubation for 15 min on ice, a clarifying spin was performed (20 min at 15,000 × g, 4°C). To the cleared lysates 1.5 µl Native PAGE 5% G-250 Sample Additive (Life Technologies, Carlsbad, CA, USA) was added and the samples were

subjected to BN-PAGE (Native PAGE 3–12% Bis-Tris; Life Technologies). After blotting on PVDF membranes (Roth, Karlsruhe, Germany) immunoblotting was performed.

## Lipid analysis

Lipid extractions were performed as described (*Bligh and Dyer, 1959*) in the presence of an internal CL standard (CL 56:0, Avanti Polar Lipids, Alabaster, AL, USA). The organic phase was evaporated and dried lipids were dissolved in 50 μl 10 mM ammonium acetate in methanol. Aliquots were diluted 1:2 with 0.1% piperidine in methanol. Mass spectrometric analysis of CL species was performed in negative ion mode on a quadrupole time-of-flight mass spectrometer (QStar Elite, AB Sciex, Darmstadt, Germany). Direct infusion was performed via an automated Chip-based system (Triversa Nanomate, Advion Biosciences, Harlow, United Kingdom). Ionization voltage was set to −0.95 kV, gas pressure to 0.5 psi. CLs were detected as $[M-H]^-$ molecules. CL species (all combinations of fatty acids 16:0, 16:1, 16:2, 18:0, 18:1 and 18:2) were analyzed by targeted product ion scanning. Data processing was performed as described (*Osman et al., 2009*).

## Acknowledgements

WN is grateful to the Max Planck Society for granting a Max Planck Senior Fellowship and to Carl Friedrich von Siemens Foundation for financial support. TI acknowledges the support by the Japan Society for the Promotion of Science Postdoctoral Fellowships for Research Abroad.

## Additional information

### Funding

| Funder | Author |
| --- | --- |
| Max Planck Society | Walter Neupert |
| Carl Friedrich von Siemens Stiftung | Walter Neupert |
| Japan Society for the Promotion of Science Postdoctoral Fellowships for Research Abroad | Toshiaki Izawa |

The funders had no role in study design, data collection and interpretation, or the decision to submit the work for publication.

### Author contributions

MEH, Conception and design, Acquisition of data, Analysis and interpretation of data, Drafting or revising the article; A-KU, TI, CÖ, SG, BB, Acquisition of data, Analysis and interpretation of data; DMW, FR, BW, Acquisition of data, Analysis and interpretation of data, Drafting or revising the article; MM, Conception and design, Analysis and interpretation of data; WN, Conception and design, Analysis and interpretation of data, Drafting or revising the article

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
