## [Decision Letter]

Thank you for sending your work entitled “Aim24 and MICOS control respiratory function, tafazzin-related cardiolipin modification and architecture of mitochondria” for consideration at *eLife*. Your article has been favorably evaluated by a Senior editor and 3 reviewers, one of whom is a member of our Board of Reviewing Editors.

The Reviewing editor and the other reviewers discussed their comments before we reached this decision, and the Reviewing editor has assembled the following comments to help you prepare a revised submission.

Harner and colleagues report that the mitochondrial protein Aim24 participates in either the maintenance of, or as a new component of, the MICOS complex that regulates the junctions between the inner and outer mitochondrial membranes. Stability of the cardiolipin transacylase tafazzin requires MICOS activity. This work expands our understanding of the emerging topic of cristae morphogenesis. But how Aim24 functions to maintain the MICOS complex remains unclear.

There are a number of important technical concerns that would need to be addressed before this work would be acceptable for *eLife*. Those issues that are most important and noted by two or more reviewers include:

1) The authors should assess the change in the amount of Mcs10 in the intact MICOS complex by co-IP experiments.

2) All three reviewers feel that the immuno EM images presented in Figure 2 do not justify the conclusion. This needs to be repeated with the contrast high enough to allow cristae junction identification and with more gold particles displayed.

3) One of the major discussion points is that loss of Aim24 together with His-tagging of Mcs12/Mcs29 destabilizes tafazzin. Is this simply a consequence of the loss of cristae junctions? What happens to tafazzin stability when for example Fcj1, Mcs10 or other components of the MICOS complex are deleted? Is Aim24 specifically involved in tafazzin stability or is it simply that cristae junctions are required?

4) Most of the results link Aim24 to MICOS indirectly. One important result is the demonstration of direct binding of Mcs10 to Aim24 in Triton X-100. Later in the manuscript the authors show synthetic phenotypes between loss of Aim24 and His-tagging of Mcs12/Mcs29. Does expression of His-tagged Mcs12 or Mcs29 affect Mcs10 binding to Aim24?

5) Given the potential problems with tags in this system, there is concern about interactions with a tagged version of Aim24 (i.e., Figure 3 in which Aim24 associates differently in Triton vs digitonin; is Aim24 tagged?). In Figure 2, it is not clear why a tagged version of Aim24-3XHA is used when an antibody is available.

---

## [Author Response]

We thank all three referees for reviewing this manuscript. Before answering their comments, we report that very recently the community working on the MICOS complex has reached an agreement on a uniform nomenclature of the MICOS complex and the genes and proteins of its subunits. This agreement, signed by 26 authors, has been submitted as a Comment to The Journal of Cell Biology and is expected to be published shortly. We are now using this nomenclature in our revised manuscript.

Please note: In our response to the comments of the referees we are still using the old nomenclature to avoid confusion.

*1) The authors should assess the change in the amount of Mcs10 in the intact MICOS complex by co-IP experiments*.

We have now quantified the levels of the MICOS subunits in the Aim24 deletion strain and in the Aim24-3HA strain relative to wild type. We have also quantified the distribution of Mcs10 in the 1.5 MDa and 300k Da fractions of gel filtration of wild type, the Aim24 deletion strain and the Aim24-3HA strain. Taking into account the increase of Mcs10 in the Aim24 deletion strain, it is clear that the increase of Mcs10 in the 300 kDa fraction cannot be the result of the increase of Mcs10 in whole mitochondria. If Mcs10 was present as an unassembled, supernumerary subunit, the subunits Mcs12, Mcs19, Mcs27 and Mcs29 which are increased like Mcs10 would be expected to be present as supernumerary species. This is not the case. Moreover, in the Aim24-3HA strain Mcs10 is not increased, but Mcs10 is completely dissociated upon gel filtration. This strongly supports the notion that indeed the deletion of Aim24, like the His-tagging of Aim24 alters the interaction of Mcs10 with the other MICOS subunits and affects the integrity of MICOS.

We have not performed co-immunoprecipitation because this procedure leads to partial falling apart of the MICOS complex. The MICOS complex is not held together by strong forces and binding of antibodies might lead to displacement of other subunits.

We have now included the determination of the steady state levels of MICOS subunits in Figure 2 and have discussed the whole issue in the text.

*2) All three reviewers feel that the immuno EM images presented in*
Figure 2
*do not justify the conclusion. This needs to be repeated with the contrast high enough to allow cristae junction identification and with more gold particles displayed*.

Unfortunately, the immuno EM image presented had lost its contrast during processing of the manuscript. We are now showing the same image with higher contrast, but have included in Figure 2 several further images demonstrating the presence of Mcs10 at crista junctions. We have also quantified the distribution of Mcs10 by evaluating 70 mitochondrial profiles and include the numbers (very similar percentage of Mcs10 in the mutant as in wild type) now in the text. We could provide the original images if there is still a loss of contrast due to conversion to PDF.

*3) One of the major discussion points is that loss of Aim24 together with His-tagging of Mcs12/Mcs29 destabilizes tafazzin. Is this simply a consequence of the loss of cristae junctions? What happens to tafazzin stability when for example Fcj1, Mcs10 or other components of the MICOS complex are deleted? Is Aim24 specifically involved in tafazzin stability or is it simply that cristae*
*junctions are required?*

We show now in Figure 6 that in mitochondria lacking Fcj1 or Mcs10 the steady state levels of tafazzin are not altered. Notably, crista junctions in these two strains are virtually absent. Therefore, loss of tafazzin is not a simple consequence of loss of cristae junctions. Also, in the absence of Aim24 tafazzin is present (see Figure 6).

*4) Most of the results link Aim24 to MICOS indirectly. One important result is the demonstration of direct binding of Mcs10 to Aim24 in Triton X-100. Later in the manuscript the authors show synthetic phenotypes between loss of Aim24 and His-tagging of Mcs12/Mcs29. Does expression of His-tagged Mcs12 or Mcs29 affect Mcs10*
*binding to Aim24?*

We generated the strains expressing Mcs10-3xHA in the wild type background, in Mcs12-His6 background and Mcs29-His6 background. Unfortunately it is not possible to co-isolate Aim24 together with Mcs10-3xHA by co-IP experiments. It seems, that the anti-HA antibody displaces Aim24.

*5) Given the potential problems with tags in this system, there is concern about interactions with a tagged version of Aim24 (i.e.,*
Figure 3
*in which Aim24 associates differently in Triton vs digitonin; is Aim24 tagged?). In*
Figure 2*, it is not clear why a tagged version of Aim24-3XHA is used when an antibody is available*.

In Figure 3 Aim24 is not tagged. In Figure 2 the HA-tagged version of Aim24 was used to demonstrate that its expression has a similar effect on the integrity of the MICOS complex upon gel filtration as deletion of Aim24, not for reasons of usage of the anti-HA antibody.